# A Cross-Sectional Study of Knowledge on Ownership, Zoonoses and Practices among Pet Owners in Northern Portugal

**DOI:** 10.3390/ani11123543

**Published:** 2021-12-13

**Authors:** Beatriz do Vale, Ana Patrícia Lopes, Maria da Conceição Fontes, Mário Silvestre, Luís Cardoso, Ana Cláudia Coelho

**Affiliations:** 1Department of Veterinary Sciences, School of Agrarian and Veterinary Sciences (ECAV), University of Trás-os-Montes e Alto Douro (UTAD), 5000-801 Vila Real, Portugal; beatrizvale.13@hotmail.com (B.d.V.); aplopes@utad.pt (A.P.L.); mcfontes@utad.pt (M.d.C.F.); lcardoso@utad.pt (L.C.); 2Animal and Veterinary Research Centre, University of Trás-os-Montes e Alto Douro, 5000-801 Vila Real, Portugal; asilvest@utad.pt; 3Department of Zootechnics, School of Agrarian and Veterinary Sciences, University of Trás-os-Montes e Alto Douro, 5000-801 Vila Real, Portugal

**Keywords:** knowledge, One Health, pet ownership, pets, Portugal, public health, zoonoses

## Abstract

**Simple Summary:**

Pet ownership is common in most countries, but few published studies have addressed pet owners’ knowledge of zoonoses, pet contact practices, or awareness of zoonotic disease risks posed by pets. The aim of this study was to assess household knowledge, attitudes, and risks related to pet ownership and zoonoses in northern Portugal. A questionnaire was developed to gather information regarding participants’ demographic characteristics; household pet types and their importance to the family; pet contact-related attitudes; knowledge of zoonoses, high-risk groups, disease transmission pathways, and disease protection measures. It was observed that most participants considered pets an important part of the family. Nevertheless, high-risk practices were recurrent and pet owners’ knowledge was limited. These results reinforce the importance of further studies to better understand the existing gaps in knowledge of pet ownership and zoonoses and strengthens the need to adopt the One Health concept.

**Abstract:**

Pet ownership is common in modern society. In Portugal, 38% and 31% of all households own at least one dog or cat, respectively. Few studies have ascertained the knowledge of pet owners on pet ownership and zoonoses, and none have been carried out in Portugal. The aim of the present study was to assess household knowledge and practices related to pet ownership and zoonoses in northern Portugal. A face-to-face questionnaire was completed by 424 pet owners, from November 2019 to February 2020. Most respondents (97.2%) considered pets as an important part of the family, especially women (*p* = 0.036); 73.1% allowed their pets to live an indoor/outdoor life; 41.3% denied sharing the bed with their pets while 29% assumed they did it daily; 20.3% reported never kissing their pets/pets licking their faces. Furthermore, 73.6% considered animals as potential sources of human diseases, but only 25.9% reported knowing the definition of zoonoses; 96.9% considered the role of veterinarians important in protecting public health. The low level of knowledge of pet owners and the occurrence of high-risk behaviors indicates a need to strengthen communication between veterinarians, physicians, pet owners, and the general public towards reduce the risk of acquisition and transmission of zoonoses.

## 1. Introduction

Pet ownership is common in modern society, although differences exist on a continent and country basis [1,2,3,4,5,6,7,8]. Dogs and cats are the most frequently owned pets although other species are gaining more attention in households worldwide [9]. According to the European Pet Food Industry (FEDIAF), 85 million European Union households (38% of all households) owned at least one pet animal in 2019. Cats represented the most commonly owned pet, followed by dogs, birds, small mammals, ornamental fishes, and reptiles. In Portugal, 38% and 31% of all households own at least one dog or cat, respectively equating to around 2,050,000 owned dogs and 1,500,000 owned cats [8].

Several studies have reported that animal ownership or interaction with animals may contribute to improve overall quality of life including physical, social and psychological health [10,11,12,13]. Nevertheless, any health benefits must be evaluated considering several variables, including age, gender, and socioeconomic status, but also the time spent with each pet and the bond that is fostered between owners and their pets, which is naturally different between members of the same household [12]. Despite the benefits, there are potential hazards associated with pet ownership and interaction with animals. Rabbits and rodent ownership during pregnancy and childhood seems to be linked to increased risk of non-atopic asthma [14,15]. Some studies report that early exposure to dogs and cats at home can predispose to pet allergies [16]. Allergies to dogs and cats affect 10 to 20% of the population worldwide and are a major risk factor in the development of allergic rhinitis and asthma [17,18,19]. Animal bites are extremely common and a serious public health problem [12,20], with dogs leading the way in humans, especially children [21]. Bites and scratches may lead to complex injuries, both physically and psychologically but rarely resulting in death. Low-income countries, where dog-control laws are seldom decreed or enforced, are estimated to have a higher incidence of dog bites, scratches and consequent fatalities due to the prevalence of rabies and poor access to appropriate healthcare [21].

The number of emerging infections from companion animals is low [22]. However, there is potential for the transmission of infectious pathogens from pets to humans (Figure 1) through skin and mucous membranes via bites, scratches, or other direct contact as well as via animal saliva, urine, and other body fluids/secretions. Additionally, the ingestion of food or drink contaminated with animal fecal material, inhalation of infectious aerosols or droplets, and exposure through arthropods or other invertebrate vectors have also been implicated [23,24].

Zoonotic agents can infect healthy people, for example through occupational exposure. In fact, occupational zoonoses frequently occur through close contact between animals and humans due to specific settings and professional activities [25]. Therefore, veterinarians, abattoir workers, researchers, and technicians, among other professionals who handle infected animals and cultures of zoonotic agents daily, can be exposed to pathogenic agents [25,26]. However, the risk of becoming infected with a zoonotic pathogen is particularly higher for those with a compromised or incompletely developed immune system such as neonates, toddlers/children (<5 years), elderly (>65 years), pregnant women and those people who suffer from primary/congenital or acquired immunodeficiencies. This includes acquired immunodeficiency syndrome (AIDS), impaired immunity from cancer, chemotherapy and radiation therapy, splenectomy, diabetes mellitus, bone marrow diseases, and consequent immunosuppressive therapy for transplants and protein-calorie malnutrition [20,24,27].

It is unrealistic to eliminate the possibility of acquiring a zoonotic disease but rather the aim should be to reduce the risk. To this end, personal hygiene measures should be applied, such as hand washing, proper animal handling, diet and health care, as well as educational measures and awareness on zoonoses prevention for pet owners, children and immunocompromised people. This, helps them make informed choices and, hopefully, interact safely with animals [12,23,28].

The One Health concept has become an international standard for expanding interdisciplinary collaboration between professionals in all aspects of human, animal, and environmental health, working at the local, regional, national and global levels [29]. The adoption of a One Health strategy and mentality is imperative to predict, prevent, detect, respond and control infectious diseases [30]. Therefore, engaging communities (pet owners and general population) through education and awareness concerning zoonotic diseases and their prevention is a fundamental part of this concept to improve human and animal health and welfare [31,32].

To date, few studies have assessed basic knowledge of pet-associated zoonoses among pet owners and there are few longitudinal studies detailing this issue [20,33,34,35,36,37,38]. Additionally, most published studies contain sparse data. More questionnaires are therefore needed with a larger sample on a region and country basis to obtain more reliable and real information on the general public’s household knowledge, attitudes, habits, and risks associated with pet ownership and animal contact. The aim of this study was to contribute to the aforementioned goal, using households with pets in northern Portugal as an epidemiological unit. Moreover, this study aimed to strengthen the importance of combating zoonoses by promoting awareness among pet owners and the general public as well as underlining the importance of animal care and raising consciousness of the reality of abandonment of companion animals, often associated with ignorance on the topic [39,40].

## 2. Materials and Methods

A cross-sectional voluntary study on zoonoses and pet ownership was conducted from November 2019 to February 2020 among a convenience sample of pet owners in northern Portugal (in the districts of Vila Real, Braga and Oporto). Portugal is a medium-sized country on a European scale and the northern region is its most densely populated area. It represents 35% of the Portuguese population which corresponds to around 3.6 M citizens, 22% of the total area of continental Portugal, and 30% of the municipalities in mainland Portugal [41]. Individuals were eligible to participate if they were at least 18 years old and had pets in their household at the time of the survey. After agreeing to participate in the study, a single member of each household was asked to complete an anonymous, confidential 15-minute self-administered written questionnaire on-site. The sample size of this study was calculated according to the formula of survey sample size [42]. Assuming a 50% default prevalence, a 95% confidence level, and a 10% absolute error, 384 participants were required for the study. To allow for a 10% non-response, the smallest sample size required was 422 participants. The questionnaire was based on a literature review [20] and designed by the authors (veterinarians and epidemiologists) to obtain information about zoonoses and pet ownership. The questionnaire consisted of closed or short answer (which can be categorized) questions which was previously pre-tested to improve validity and reliability. Thereafter, it was applied to 50 non-participant pet owners before the study was initiated to help improve clarity of the questions and ensure that the estimated time needed to complete the survey (approximately 15 min) was accurate. The questionnaire gathered both individual and household-level data including respondents’ demographic characteristics (gender, age, education level, residence and occupation) while definitions of urban/rural residence and physical/mental occupation were not present in the questionnaire but rather based on respondent’s definition. Animal data included: (1) type of pets in the household and their importance to the family; (2) pet and animal contact-related attitudes (provide basic care-food, shelter; take the animal to the veterinarian; communicate to the responsible entities or to non-governmental entities; not take any action); (3) knowledge of the term “zoonoses” and if so, examples of zoonotic diseases and sources for such information; (4) existence of high-risk groups; (5) disease transmission pathways; (6) disease protection measures and their importance; and (7) the occurrence of animal contact and pet-associated zoonotic disease and injury. In the questions with closed-ended responses on the importance of pets in the family and of protective measures, a 5-point scale answer ranked from 1 to 5 (1 = very little important; 2 = little important; 3 = important; 4 = very important; 5 = extremely important) was used. In the question with closed-ended responses on the importance of veterinarians in protecting public health, a 4-point scale answer ranked from 1 to 4 (1 = minimum importance; 2 = some importance; 3 = very important; 4 = maximum importance) was used. In the questions with closed-ended responses on pet owners’ knowledge of zoonoses and protective measures, a 5-point scale answer ranked from 1- to 5 (1 = no knowledge; 2 = little knowledge; 3 = some knowledge; 4 = a lot of knowledge; 5 = excellent knowledge) was used.

### Data Analysis

Data were entered into an Excel database (Microsoft Corp., Redmond, WA, USA) and exported and analyzed using SPSS version 27.0 (SPSS, IBM Corporation, New York, NY, USA). For descriptive purposes, the Pearson χ^2^ test was conducted for each variable in the study looking at gender and ownership differences. The *t*-test was conducted in order to examine the owners’ gender differences relative to the importance attributed to knowledge and practices regarding pet animals and zoonoses. Statistical significance was based on a *p*-value < 0.05.

## 3. Results

In the period in which the survey was applied, 424 individuals responded. Dogs were the most frequently reported owned species (72.9%; *n* = 309), followed by cats (52.8%; *n* = 224), birds (9.6%; *n* = 41), turtles (5.4%; *n* = 23), fish (2.6%; *n* = 11), rabbits (1.9%; *n* =8), exotic animals (1.2%; *n* = 5), horses (0.7%; *n* = 3), and livestock species (0.2%; *n* = 1). Cats and dogs were owned by 97.6% of respondents (414/424).

### 3.1. Demographics

Animal owners had an average age of 33 years. The youngest respondent was 18 years old and the oldest 80 with 54% of respondents being young adults (≤35 years of age) and 46% adults (>35 years of age). Regarding gender, 74.1% (*n* = 314) were female and 25.9% (*n* = 110) male. In relation to academic background, 49.5% (*n* = 210) had higher education (attended or completed), 25.5% (*n* = 108) secondary education (up to the 12th school year), 10.6% (*n* = 45) the 3rd cycle of basic education (up to the 9th school year), 5.9% (*n* = 25) the 2nd cycle of basic education (up to the 6th school year), and 8.5% (*n* = 36) the 1st cycle of basic education (up to the 4th school year). Regarding occupation, 28.5% (*n* = 121) were students, 39.9% (*n* = 169) had an essentially physical professional activity, 19.1% (*n* = 81) an essentially mental activity, and 12.5% (*n* = 53) no occupation (unemployed or retired). Most respondents lived in urban areas (58.5%; *n* = 248) and 41.5% (*n* = 176) rural areas (Table 1).

### 3.2. Opinion on the Importance and Benefits of Having Pets

Most respondents considered pets as an important part of the family (97.2%; *n* = 412), with most individuals of this opinion being dog owners (73.1%; *n* = 301).

Participants (*n* = 412) were asked to assign a level of importance that their animals constituted for their family, where 1 was the minimum level and 5 the maximum. Most respondents considered them extremely important (level 5; 43.4%; *n* = 184) or very important (level 4; 30.2%; *n* = 128). Level 3 of importance was chosen by 20.3% (*n* = 86) of participants, followed by level 2 (2.1%; *n* = 9) and level 1 (1.2%; *n* = 5), which correspond to little importance and very little importance, respectively.

In this study, 75.2% (*n* = 319) of the participants considered that benefits of having a pet outweighed the potential health risks. This opinion prevailed among women (χ^2^ = 6.082; *p* = 0.048) and younger respondents (χ^2^ = 23.498; *p* = 0.000), while 16.7% (*n* = 71) of all respondents had no opinion.

Most participants, (76.4%; *n* = 324) considered that having no pets (or reducing the number of pets in the household) would negatively affect their family. In contrast, 8.5% expressed no opinion.

### 3.3. Pet and Animal Contact-Related Attitudes

It was reported that 73.1% (*n* = 310) of the companion animals lived an indoor/outdoor life; 79.7% of participants stated that their animals never ate or licked the dishes, in contrast to 2.1% of the respondents who reported that this occurred daily. In addition, 24.1% of owners admitted washing the pet food containers together with other dishes (Table 2).

In this study, 41.3% of participants denied that they or someone in the family shared their bed with their pets and 29% assumed they did so constantly. The responses “frequently” and “always” were those chosen most by dog owners (χ^2^ = 13.351; *p* = 0.004). Moreover, 20.3% of respondents reported that they never kissed their pets, or their animals never licked their faces. However, the majority admitted doing this, particularly among female owners (χ^2^ = 14.537; *p* = 0.002) (Table 2).

In relation to contact with other animals, 54.5% (*n* = 231) of owners declared that their pets had contact with other animals. About half of the owners who answered affirmatively (50.6%; *n* = 117) considered that these animals could be a risk for the acquisition of diseases.

### 3.4. Attitude towards Stray Animals

When asked about their attitude towards a dog or cat that they frequently found close to home, 43.4% (*n* = 184) of participants answered that they would provide food and shelter. Respondents who did not have dogs at home reported having this attitude more frequently (χ^2^ = 3.995; *p* = 0.046) as well as female respondents (χ^2^ = 4.808; *p* = 0.028). Only 7.1% (*n* = 30) would take the animals to the vet to assess their health status for deworming and/or vaccination. Moreover, 37.3% (*n* = 158) would communicate to the responsible entities (city council, municipal kennel) or to non-governmental entities (veterinary centers, animal protection associations). This attitude was statistically associated with having a dog (χ^2^ = 5.081; *p* = 0.024). In this study, 31.3% of the participants would not act, which implies that they would not help the animal or try to resolve the situation. Most of those who would not act were the older respondents (χ^2^ = 10.381; *p* = 0.001).

### 3.5. Opinion Regarding Stray Dogs

The majority (85.4%; *n* = 362) of the respondents considered that stray dogs should be collected and treated (surgical and/or medical treatment should be applied when necessary). Most participants who had this opinion were female (χ^2^ = 7.243; *p* = 0.007). Only 1.2% (*n* = 5) considered that stray dogs should be euthanized because of their potential for disease transmission. For 6.4% of respondents, stray dogs should be left freely on the streets if they were first neutered, vaccinated, and dewormed. The remaining 7% had no opinion.

### 3.6. Visits to the Veterinarian

Participants were asked to state in what situations they took their animal to the vet within a list of options. Most owners (93.2%; *n* = 395) took their pet to the veterinarian while 6.8% (*n* = 29) did not. Most reported taking their animals to the vet for vaccination or deworming (79.2%; *n* = 336) or due to illness (74.8%; *n* = 317), particularly female respondents (χ^2^ = 4.418; *p* = 0.036). More than half did not take their animals to routine consultations (64.9%; *n* = 275). Only 6.8% (*n* = 29) declared going to the vet for guidance on issues related to estrus, pregnancy, and pet-offspring. In addition, only 13.2% of participants answered taking their pets for bathing and shearing. Four participants (0.9%) reported other motives, such as cutting nails, bee stings, and oral hygiene procedures, such as scaling and trimming rabbits’ teeth.

### 3.7. Deworming

Most participants reported internally deworming their pets (90.6%; *n* = 384). Of these, 44.1% (*n* = 187) dewormed their animals every six months; 26.6% dewormed frequently than every six months; 20.3% (*n* = 86) annually, and 9.4% (*n* = 40) declared not doing so.

Regarding external application of antiparasitic drugs, 93.9% (*n* = 398) of participants reported protecting their animals against external parasites. Of these, 37.5% (*n* = 159) performed prevention for external parasites every six months; 42.5% (*n* = 180) applied antiparasitic drugs for external parasites more often than every six months; 13.9% (*n* = 59) prevented external parasites annually, and 6.1% (*n* = 26) did not perform preventive measures for external parasites.

### 3.8. Zoonotic Disease Knowledge and Educational Sources

More than half of respondents (58.7%; *n* = 249) were comfortable with their level of knowledge in relation to the potential diseases that may arise from contact with animals, which prevailed among the younger respondents (χ^2^ = 6.140; *p* = 0.013).

Participants were asked to assign their perception of the level of their knowledge on this topic. Most participants reported having no knowledge (40.8%; *n* = 173) followed by respondents who reported having some knowledge (36.3%; *n* = 154), little knowledge (11.3%; *n* = 48), a lot of knowledge (9.7%; *n* = 41) and excellent knowledge (1.9%; *n* = 8).

Most participants (73.6%; *n* = 312) considered animals as potential sources of human diseases. However, 8.5% (*n* = 36) answered negatively and 17.9% (*n* = 76) admitted not knowing. Interestingly, only 25.9% (*n* = 110) of respondents reported knowing the definition of zoonoses while in contrast, 74.1% (*n* = 314) acknowledged not being aware of this concept.

Examples of zoonoses were requested. The frequencies of the responses were as follows: rabies (13.2%; *n* = 56), brucellosis (7.5%; *n* = 32), toxoplasmosis (7.3%; *n* = 31), leishmaniosis (5.0%; *n* = 21), scabies (5.0%; *n* = 21), dermatophytosis (4.5%; *n* = 19), tuberculosis (2.8%; *n* = 12), salmonellosis (2.1%; *n* = 9); leptospirosis (1.9%; *n* = 8), babesiosis (1.2%; *n* = 5), tick fever (0.9%; *n* = 4), dengue (0.7%; *n* = 3), giardiosis (0.7%; *n* = 3), yellow fever (0.7%; *n* = 3), avian chlamydiosis (0.5%; *n* = 2), bovine spongiform encephalopathy (0.5%; *n* = 2), colibacillosis (0.5%; *n* = 2), and malaria (0.5%; *n* = 2). Others included Q fever (0.5%; *n* = 2), taeniosis (0.5%; *n* = 2), West Nile disease (0.5%; *n* = 2), avian influenza (0.2%; *n* = 1), COVID-19 (0.2%; *n* = 1), cryptococcosis (0.2%; *n* = 1), Ebola (0.2%; *n* = 1), ehrlichiosis (0.2%; *n* = 1), heartworm (0.2%; *n* = 1), hydatidosis/echinococcosis (0.2%; *n* = 1), Lyme disease (0.2%; *n* = 1), rickettsiosis (0.2%; *n* = 1) and trichinellosis (0.2%; *n* = 1).

Participants were asked how they obtained information about zoonoses. For 20.3% (*n* = 86), this was through academia followed by conversations with veterinarians (9.2%; *n* = 39), media (4.2%; *n* = 18), or family and friends (4.0%; *n* = 17), with only 0.9% (*n* = 4) through physicians.

### 3.9. High-Risk Individuals

Most participants (68.4%; *n* = 290) were aware of groups of people who are at greater risk of contracting zoonotic diseases, while 22.2% (*n* = 94) admitted not knowing what to answer, and 9.4% (*n* = 40) considered that there are no high-risk individuals. The degree of risk assigned is shown in Table 3.

### 3.10. Transmission of Zoonotic Diseases

Participants were asked if they knew how animals could transmit diseases to humans and the majority responded affirmatively (64.6%; *n* = 274). Of these participants, 75.9% (*n* = 208) considered feces as a means of transmitting zoonotic diseases, and this option was more frequently chosen by female owners (χ^2^ = 3.964; *p* = 0.046) while 64.2% (*n* = 176) considered that transmission could occur through insect bites with female owners again more frequently selecting this option (χ^2^ = 4.805; *p* = 0.028).The option “blood” was chosen by 56.7% (*n* = 155) of the respondents, followed by “physical contact” (48.9%; *n* = 134) while 48.2% (*n* = 132) considered that transmission could occur through coughing and sneezing with male owners choosing this option more frequently (χ^2^ = 5.302; *p* = 0.021). Additionally, 43.1% (*n* = 118) recognized the food route as a means of transmitting zoonotic diseases, followed by vomit (41.6%; *n* = 114) and 38.3% (*n* = 132) deemed it possible to transmit zoonotic diseases through fur, brushes and blankets of the animals.

### 3.11. Knowledge of Preventivee Measures

Participants were asked if they felt comfortable with their knowledge in relation to preventive measures against zoonotic diseases and 52.1% (*n* = 221) responded affirmatively. Subsequently, the respondents had to assign the level of knowledge they thought they had. The majority admitted not having knowledge (level 1; 47.9%; *n* = 203) followed by having some knowledge (level 3; 35.6%; *n* = 151), little knowledge (level 2; 8.7%; *n* = 37), a lot of knowledge (level 4; 6.1%; *n* = 26) and excellent knowledge (level 5; 1.7%; *n* = 7).

Respondents were asked to attach a level of importance to general protection measures. The highest level of importance (level 5) was assigned to primary hygiene care (41.5%), followed by frequent hand washing (38.9%) and people vaccination (33.3%) (Table 4).

Subsequently, the participants who considered themselves comfortable with their level of knowledge, had to assign a level of importance for preventive measures directly associated with animals. Female owners attributed a higher level of importance for two preventive measures: treatment of sick animals (χ^2^ = 13.545; *p* = 0.019) and not feeding raw meat to pets (χ^2^ = 15.552; *p* = 0.008). The results are shown in Table 5.

### 3.12. Concern about Zoonotic Diseases

When their animals become ill, 55.7% (*n* = 236) declared being concerned about the possibility of their pets transmitting diseases to themselves or to some family member. Those who answered affirmatively were asked to attribute a level of concern and the majority admitted being concerned (level 2; 49.2%; *n* = 116), followed by those being just a little concerned (level 1; 22.9%; *n* = 54), very concerned (level 3; 19.1%; *n* = 45) and extremely concerned (level 4; 8.9%; *n* = 21).

### 3.13. Acquired Diseases by Pets and Pet Owners

Owners were questioned whether their pets ever had a disease that could be transmitted to humans. According to the results, few animals contracted zoonotic diseases (4.0%; *n* = 17), while 26.4% (*n* = 112) of the participants answered not knowing, and the majority (69.6%; *n* = 295) responded that their animals never had such diseases. Those who answered affirmatively declared their animals suffered from: dermatophytosis (*n* = 11), scabies (*n* = 2), babesiosis (*n* = 1), leishmaniasis (*n* = 1), leptospirosis (*n* = 1) and parasitism (*n* = 1).

In this survey, 2.6% (*n* = 11) of pet owners confirmed that they had some diseases transmitted by animals, which were dermatophytosis (*n* = 6), parasitism (*n* = 3), brucellosis (*n* = 1) and scabies (*n* = 1).

### 3.14. Access to Information

In this study, 87.0% (*n* = 369) of respondents would like to obtain more information about zoonotic diseases but 7.5% (*n* = 32) had no opinion.

Of the participants who answered affirmatively, 66.9% (*n* = 247) would like to have it through the media (TV, radio, social media network), 64.7% (*n* = 239) through veterinarians, 34.1% (*n* = 126) through physicians, and 33.3% (*n* = 123) through flyers.

### 3.15. The Role of Veterinarians

In this study, 96.9% (*n* = 411) of the participants considered the role of veterinarians important in protecting public health, although 3.1% (*n* = 13) had no opinion. No participant responded negatively.

Those who answered affirmatively were asked to assign a level of importance in which 1 represented minimum importance and 4 maximum importance. More than half (56.6%; *n* = 240) attributed the maximum level of importance. For 32.1% (*n* = 136), the role of veterinarians is very important (level 3), followed by the owners who attributed them some importance (level 2; 7.5%; *n* = 32) and minimum importance (level 1; 0.7%; *n* = 3). Female owners assigned the highest score values (χ^2^ = 14.792; *p* = 0.005) as well as the younger respondents (χ^2^ = 21.025; *p* = 0.000).

### 3.16. Differences in the Gender of Pet Owners Relative to the Importance Attributed to Different Questions

Regarding the mean score, the *t*-test revealed that female pet owners attributed a higher value to the importance of pets in family (*t* = 2.103; *p* = 0.036). The same scenario was seen in relation to the role of veterinarians in protecting public health (*t* = 3.94; *p* = 0.000). Table 6 shows the differences between the gender of the pet owners and the importance each one attributed to different questions.

## 4. Discussion

This study aimed to characterize household knowledge, attitudes and practices related to pet ownership in northern Portugal. Despite the considerable increase in the number of households with pets, the close interaction of animals with household members and the growing recognition of the potential benefits and risks pets posed, few studies have addressed this topic [20,33,43,44,45], and none have been conducted in Portugal. Although limited by a small sample of Portuguese pet owners, this study can be a useful tool to enlighten veterinarians, physicians, politicians, and other professionals as well as the general public about pet owners’ knowledge and practices.

### 4.1. Type of Pets, Their Benefits and Risks

Animal ownership patterns observed were consistent with the FEDIAF report on Portugal with most respondents having dogs, followed by cats and birds. Most participants considered their animals an important part of the family and believed that benefits of pet ownership outweigh disease risks. Indeed, several studies have reported cardiovascular benefits [46], especially in the elderly hypertensive and diabetic population [47,48,49,50,51]. Pet ownership during pregnancy, in the first year of life, and during childhood seem to prevent the development of food allergies [52] and reduce the risk of wheezing (associated with cat ownership) [15], atopic asthma, and aeroallergen sensitization (grass, house dust mite, cat and dog allergens) [14,53,54,55]. Dogs also increase opportunities for recreational walking, thereby reducing the risk of obesity [56,57,58,59,60]. Pet ownership or simply being in contact with animals seems to provide social support, reduce distress and anxiety and decrease loneliness and depression [10] particularly among older individuals [61], children [62,63], and adolescents with autism spectrum disorder [64], immunocompromised individuals [65], and homeless people [66].

### 4.2. Knowledge on Zoonotic Diseases

It is crucial to emphasize that few studies have ascertained the level of knowledge of pet owners about zoonoses and their prevention [20,34,43,45] and some have only focused on dog-associated zoonoses or zoonoses associated with endoparasites [37,38,67,68,69]. In addition, there are even fewer studies on this topic that address new companion animals (reptiles and small mammals) [70].

Pet owners, for the most part, were comfortable with their level of knowledge of zoonotic diseases although 40.8% admitted not having knowledge. To assess zoonotic disease knowledge, respondents were asked about the definition of zoonoses. Interestingly, most respondents did not know the meaning of the word or its definition. However, they considered animals as a potential source of diseases. This finding is in line with that of other studies, which reported a lack of knowledge of pet owners and/or general public [20,38,45,68]. Moreover, since a considerable part of the surveyed pet owners were comfortable with their knowledge, they are unlikely to seek information on the topic.

Pet owners who claimed to understand the term “zoonoses” were asked for examples. Rabies was the most frequent zoonotic disease reported, in agreement with other world studies [20,33,43]. Portugal is a country that has been officially rabies-free since 1961 and vaccination of dogs against this disease is mandatory [71]. Perhaps this is one of the reasons for the greater knowledge of owners about this disease compared with other more common diseases in the country such as salmonellosis, dermatophytosis or scabies.

Statistically significant differences existed regarding the gender of the participants who referred to toxoplasmosis as an example of zoonoses. In fact, only one male mentioned this zoonosis, which may suggest that women are more aware of this disease, perhaps because they are more informed that childbearing women constitute a risk group. However, given the small sample and the type of questionnaire performed, it is not possible to conclude this assumption. Gargaté et al. [72] referred a lack of knowledge of the current epidemiological situation in Portugal, considering that the only toxoplasmosis national serological survey was performed in 1979/1980. This study indicated that more than 80% of Portuguese pregnant women were susceptible to primary infection [72]. In addition, a seroepidemiological study developed in northern Portugal emphasized the susceptibility of childbearing women to primary infection with *T*. *gondii*. Moreover, the study reported that risk factors for *T*. *gondii* infection in women were participating in soil-related activities without gloves, consumption of unwashed raw vegetables or fruit, and consumption of smoked or cured (non-cooked) processed pork products [73].

A recent study conducted in Italy [74] which aimed to evaluate the knowledge of Italian women about toxoplasmosis concluded that this was superficial and incomplete. Similar studies must be implemented in Portugal as well as effective education and learning programs.

### 4.3. Educational Sources about Zoonoses

Most of the participants who declared knowledge of zoonoses admitted having acquired it through their academic path, contradicting other studies which reported veterinarians as the main source of zoonotic disease information for pet owners [20,28,33]. However, in both cases, the role of veterinarians as educators was neglected. In fact, there are several studies reporting the lack of transmission of information about zoonotic diseases and their potential hazards by veterinarians to animal owners [20,33,45,75,76,77]. Pet owners indicated that veterinarians only discuss this issue when asked or whenever zoonoses had been diagnosed in pets [76]. In this study, most of the respondents considered the role of veterinarians important in protecting public health. Interestingly, the majority would like to obtain information about zoonotic diseases through the media. Veterinarians were the second most chosen category. Although the difference between them is not significant, it might be prudent to rethink the way in which information is transmitted. Providing client educational materials on zoonotic diseases could be interesting [78] as well as reliable resources online [40]. Nevertheless, veterinarians will always play a central role in the promotion of pet owners’ education about zoonotic diseases [28].

Findings from this study are consistent with those of others regarding the role of physicians in transmitting information about zoonoses. In fact, the limited involvement of physicians in questioning the detention of pets by their patients, advising on protective measures and discussing potential zoonotic diseases has been described [20,28,45,77,79,80,81]. This is problematic since animal contact frequently occurs in both pet and non-pet owning households, so individuals without pets but with animal contact will likely not consult a veterinarian. Moreover, veterinarians are often unaware of the health status of the pet owners and their household members [20,78]. Therefore, physicians are the most suitable health professionals to advise and inform patients on their household zoonotic disease risks [20]. Nevertheless, veterinarians should discreetly question their clients to ascertain whether they or other household members are especially vulnerable because of pregnancy, age or immunosuppression and counsel them about appropriate precautions [45].

The One Health concept is in vogue. It is essential that the veterinary and human fields join efforts to bridge the communication and information gaps that have been reported over the years [28]. These failures are felt not only between physician/patient and veterinarian/pet owner but also among professionals in both areas. The gap between the two professions begins even during their early professional education and training [28,82], since medical and veterinary students are rarely provided with opportunities for inter-professional learning and collaboration [83]. The One Health concept foresees using their knowledge and skills to enhance the clinical management of zoonoses in humans and animals which will ultimately benefit human, animal, and environmental health [28,82,83].

### 4.4. Pet Related Practices

The growing importance of pets, especially in industrialized countries, has led to increasingly close contact with their owners [84]. Some studies have shown that between 13% and 63% of owners allowed their pets to sleep on their bed and 40–60% of pets were allowed to lick their owner’s face [13,34,38,45,68,84,85,86]. The results of the present study are consistent with these observations. Zoonotic infections acquired by sleeping with a pet or by being licked on the face are uncommon. However, these behaviors can increase pet-associated disease risks. Bacterial infections, such as *Yersinia pestis* (plague’s agent), *Bartonella henselae* (cat scratch disease), Methicillin-resistant *Staphylococcus aureus* [84] and a few cases of fatal bite wound infections by *Capnocytophaga canimorsus* [87] and *Pasteurella multocida* [88] have been documented. Zoonotic transmission through this route for other pathogens such as gastric *Helicobacter* spp. [89] and periodontal pathogens [90] have been reported.

Apart from the abovementioned risk factor, more than 70% of respondents admitted that their pets had free access to the interior of the house and could roam indoors. Some participants reported household husbandry practices that increase zoonotic disease risk, such as allowing their pets to eat or lick their dishes, washing pet food containers with other dishes, or allowing their pets to be in contact with potentially disease-carrying animals. Other studies reported these practices, but also high-risk habits, such as feeding their animals with raw meat, eggs, and animal product treats, washing their pets in the kitchen sink, and removing pet feces from the garden/backyard/litter box weekly or less often [34,40,91]. In Portugal, 10.1% of pets are fed homemade or alternative diets which are often nutritionally unbalanced [40]. Furthermore, raw animal products in pet diets are a well-established risk factor for salmonellosis in dogs [34,91,92,93], cats [93,94,95] and humans. Although human salmonellosis derived mainly from handling or consuming contaminated food products, it can occur due to contact with companion animals, natural pet treats and biological waste from pets consuming raw diets [92,96,97,98]. Moreover, reports of raw meat pet food containing other zoonotic foodborne bacteria such as *Escherichia coli* and *Listeria monocytogenes* are increasing [98,99]. Pet owners can reduce this public health risk by not feeding natural pet treats and raw food diets to their pets [97,98]. It should also be noted that reptiles are becoming increasingly popular as pets and constitute reservoirs of a wide variety of *Salmonella* serotypes [70]. The high prevalence of *Salmonella* shedding by healthy reptiles and high incidence of human salmonellosis attributed to contact with reptiles have been described, especially among children younger than five years old [70,100,101,102,103,104]. Therefore, safe reptile handling recommendations must be conveyed to pet owners by physicians, veterinarians, public health professionals, and industry officials through conversations, educational materials, and interventions [34,70,102].

Overall, more than 90% of respondents took their pets to the veterinarian with the most common reasons being for vaccination, antiparasitic preventive treatment or health-issues. A similar scenario was reported in other studies [6,34,40,105,106]. It is noteworthy that this study did not explore the association between demographics, animal’s characteristics, or socioeconomic factors.

Most participants declared that their pets were annually subjected to two to four preventive treatments for gastrointestinal parasites and external parasites. A study developed in Portugal by Prata [40] showed equivalent results while highlighting a considerable number of participants stating that their pets were submitted to antiparasitic preventive treatment once a year. Concerning deworming practices by Portuguese pet owners, similar findings were concluded by Pereira et al. [37]. In contrast, Matos et al. [68] showed a considerably high percentage for monthly ectoparasite treatments (dogs: 50.5%; cats: 17.2%). Pereira et al. [37] assumed that this difference may be due to the population sampled or possible bias caused by face-to-face survey. However, the present study was based on face-to-face questionnaire and the same result was not obtained.

### 4.5. High-Risk Groups

Pet-human contact is frequent, but for most people, such contact does not translate into a high health risk situation [34]. However, more vulnerable groups exist, who can become ill after infection, such as those who are immunocompromised (HIV/AIDS patients), having had transplants, oncology patients receiving chemotherapy, post-splenectomy, chronically ill (such as diabetes patients), children (<5 years), the elderly (>65 years), and pregnant women [13,45]. Additionally, they may have more severe complications and symptoms with longer durations [34]. In this study, the presence of high-risk individuals in the households was not determined. It was only intended to assess the participants’ perception of the existence of high-risk groups. Although most respondents are aware of the existence of people who are more vulnerable to contracting zoonotic diseases, it is alarming that, for each risk group presented, 30–40% of respondents did not consider them more likely to develop severe clinical conditions after infection. This result is in line with the lack of knowledge found about zoonoses in this study and others [20,34,35].

Younger children are recognized for their hand-to-mouth behavior. Additionally, children and people with some developmental disabilities are more prone to poor hygiene care or higher risk contact with animals, which makes them particularly exposed to infection [13,35,36]. There are also situations where risk groups intersect, such as children with cancer or diabetes. A study developed by Stull et al. [35] reported that most households with immunocompromised children or children with diabetes acquired a new pet considered high-risk for infectious disease (reptiles or amphibians, rodents and exotic species). Moreover, interactions between parents and physicians/staff members about pet ownership or zoonotic disease information were generally uncommon [35]. Households with higher risk individuals are assumed to be aware of pet-associated diseases. Overall, medical staff and veterinarians must guide patients/clients in pet selection, preventive measures (such as hand hygiene) and changes in animal contact to reduce health risks [35].

### 4.6. Zoonotic Disease Transmission Pathways

When questioned about the possibility of disease transmission from animals to humans, more than half (64.6%) of the pet owners in the present study responded affirmatively. However, given a list of possible pathways of transmission, it was found that participants rarely chose all options, which proves some lack of knowledge on the topic. It was encouraging to note that most of the respondents were aware that feces, arthropods, and blood were important means for transmitting zoonotic diseases. Nonetheless, it was worrying to realize that only less than half were conscious of physical contact (scratches and bites), food route (raw meat, raw eggs, etc.) and fur as potential transmission pathways. It was previously mentioned how raw meat and close physical contact between owners and pets (sleeping with pets, allowing pets to lick the face or wounds, bite accidents, scratches) constitute an animal and public health risk. In relation to fur, both dogs and cats regularly lick the anus and thereafter the fur, increasing the odds of Enterobacteriaceae on their fur or footpads [13]. The prevalence of infectious embryonated eggs of *Toxocara* spp. on dog’s fur is low although it should not be neglected [107]. A survey conducted by Overgaauw et al. [86] reported that in 12% and 3% of investigated dogs and cats, respectively, *Toxocara* eggs were found in fur but not in feces. These results prove that the presence of eggs in fur is not always due to self-contamination. The significantly higher prevalence in dogs compared to cats may be due to their behavior: dogs tend to have greater soil contact (rolling around, digging, etc.) while cats have extreme grooming habits that lead to the removal of possible existing eggs [86].

### 4.7. Concern about Zoonotic Diseases

Contrarily to Steele and Mor [45] who reported that 63.9% of pet owners were not concerned, about 56% of respondents in the present study admitted to being worried about the possibility of their pet contracting and transmitting a zoonotic disease.

Furthermore, few (4%) in the present study reported that their animals contracted zoonotic diseases in the past. A similar answer was obtained when asked about themselves (2.6%). These results are similar to a Canadian study in which 4% of households reported getting a disease from their pet [20] and to the Australian study developed by Steele and Mor [45] in which only 2.5% of the respondents reported having caught a disease from their pet. Given the high lack of knowledge about zoonotic diseases by a considerable portion of respondents, it is expectable that this result may have been underestimated. Moreover, it is possible that respondents were not comfortable enough to admit it.

### 4.8. Preventive Measures of Zoonotic Diseases

Routine preventive veterinary care and husbandry practices are important to reduce zoonotic diseases [34,36]. When questioned about their knowledge of measures to prevent zoonotic diseases, half of the respondents considered they had some qualifications. Furthermore, the vast majority rated all preventive measures mentioned at their higher level. However, the other half of participants admitted not having enough knowledge to determine their importance, which is worrying and alarming. This result again proves the need to inform the general public about zoonoses and all their branches. The collective effort of all competent entities is essential to, for once and for all, educate people and consequently keep them and their pets safe.

Some studies [20,34,35,38,45,68] reported practices performed by participants to reduce the risk of infection and spread of zoonotic diseases. However, in the present study, the respondents were not asked about their actions towards the goal mentioned above, but rather the importance they attributed to each measure, so a direct comparison of the results is not feasible. Nonetheless, the importance of hand washing was unanimous.

### 4.9. Gender Differences

The aim of this study was to gather information on pet owners from northern Portugal. The survey was conducted with a convenience sample and estimated proportions may not be representative of all pet owners from this region. The higher response proportion by females most likely relates to their willingness to participate in paper-and-pencil surveys [108].

### 4.10. Study Limitations

In retrospect, the present study could have included questions about the existence of high-risk individuals in the households in order to understand whether this condition would interfere with knowledge and willingness to learn more about zoonotic diseases. Additionally, open questions about preventive measures could have been performed to really look at what measures respondents apply on a daily basis and finally, owners could have been asked about specific common zoonoses rather than just being asked for the definition and some examples. In this way, it would be possible to verify in depth the knowledge of the participants about the zoonoses they had referred to.

Knowledge and practices may vary by species owned. However, since cats and dogs were owned by most of the pet owners (97.6%) and other species were seldom reported, the potential bias was minimal.

## 5. Conclusions

This study revealed a lack of knowledge and understanding of zoonoses among northern Portuguese pet owners. Veterinarians and physicians must be proactive in their public health responsibilities, implementing the One Health concept and strengthening communication between both professions as well as with their clients/patients. Different methods should be used to draw people’s attention to the topic of zoonoses and their prevention. Addressing the subject during consultations is essential, but not completely effective. Free online educational programs, audiovisual presentations, and flyers can promote better educational mediums for people of different ages, academic backgrounds, and willingness to listen and assimilate the information provided. These educational vectors must contain information about the most common zoonoses, transmission pathways, risk behaviors, preventive measures (promoting routine hygiene practices, such as hand washing, cleaning up dog feces or removing feces from cat litter daily, not feeding their animals with raw meat and eggs, encouraging vaccination, and deworming of their pets, etc.) and high-risk groups (children, elderly, pregnant women and those immunocompromised). Medical and veterinary school students should be exposed to the One Health concept from the beginning of their career, improve their knowledge on current zoonotic disease risks, foster their team spirit, collaboration and interprofessional dialogue. From the scientific point of view, more questionnaires, pilot studies and cross-sectional studies should be developed to frequently update eventual progress. All the above-mentioned measures are efforts that must be taken to disseminate knowledge and, consequently, reduce pet-associated disease risks to the public.

## Figures and Tables

**Figure 1 animals-11-03543-f001:**
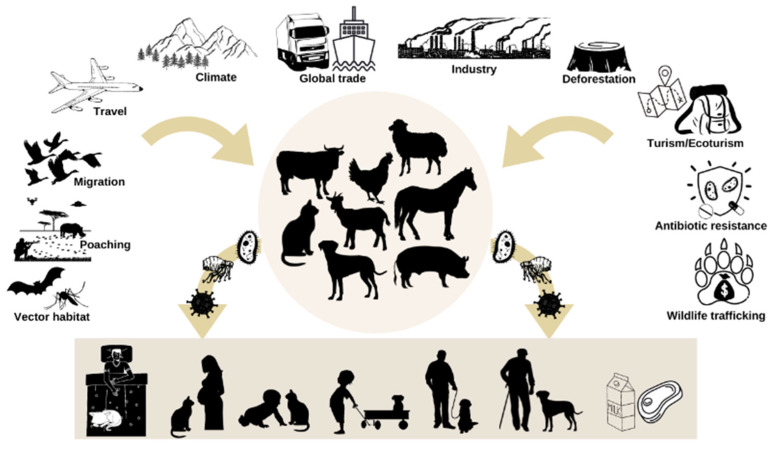
Factors responsible for the emergence of zoonoses and their transmission to humans.

**Table 1 animals-11-03543-t001:** Demographics of respondents who participated in the survey in northern Portugal in 2020 (*n* = 424).

Characteristic		*n* (%)
Gender	Female	314 (74.1)
	Male	110 (25.9)
Age (years)	Median	33
		229 (54.0)
		195 (46.0)
		60 (14.2)
		126 (29.7)
		98 (23.1)
Education	1st cycle of basic education	36 (8.5)
	2nd cycle of basic education	25 (5.9)
	3rd cycle of basic education	45 (10.6)
	Secondary school	108 (25.5)
	Higher education	210 (49.5)
Occupation	Student	121 (28.5)
	Essentially physical activity	169 (39.9)
	Essentially mental activity	81 (19.1)
	No occupation (unemployed or retired)	53 (12.5)
Residence	Urban	248 (58.5)
	Rural	176 (41.5)

**Table 2 animals-11-03543-t002:** High-risk practices for the transmission of zoonotic diseases among respondents in northern Portugal in 2020 (*n* = 424).

Practices	Never *n* (%)	Sometimes *n* (%)	Frequently *n* (%)	Daily *n* (%)
Pet licks/eats from the owner’s plate	338 (79.7)	62 (14.6)	15 (3.5)	9 (2.1)
Sharing bed with the pet	175 (41.3)	56 (13.2)	70 (16.5)	123 (29.0)
Owner kisses the pet/pet licks owner’s face ^1^	86 (20.3)	135 (31.8)	90 (21.2)	113 (26.7)

^1^*p* = 0.002. Superior among female owners.

**Table 3 animals-11-03543-t003:** Risk score attributed to each high-risk group by respondents in northern Portugal in 2020 (*n* = 424).

Risk Score ^a^	0 *n* (%)	1 *n* (%)	2 *n* (%)	3 *n* (%)	4 *n* (%)	5 *n* (%)
Elderly	161 (38.0)	26 (6.1)	24 (5.7)	65 (15.3)	86 (20.3)	62 (14.6)
Cancer patients	156 (36.8)	17 (4.0)	16 (3.8)	33 (7.8)	79 (18.6)	123 (29.0)
Pregnant women	148 (34.9)	12 (2.8)	18 (4.2)	65 (15.3)	91 (21.5)	90 (21.2)
Children	159 (37.5)	23 (5.4)	39 (9.2)	80 (18.9)	82 (19.3)	41 (9.7)
HIV/AIDS patients	171 (40.3)	25 (5.9)	20 (4.7)	50 (11.8)	53 (12.5)	105 (24.8)
Transplant patients	159 (37.5)	12 (2.8)	12 (2.8)	46 (10.8)	71 (16.7)	124 (29.2)
Chronic disease patients	172 (40.6)	41 (9.7)	52 (12.3)	73 (17.2)	59 (13.9)	27 (6.4)

^a^ 0: no risk; 1: minimum risk; 2: some risk; 3: reasonable risk; 4: too much risk; 5: maximum risk.

**Table 4 animals-11-03543-t004:** Level score attributed to the importance of each general preventive measure by respondents in northern Portugal in 2020 (*n* = 424).

Level of Importance ^1^	0 (*n*; %)	1 (*n*; %)	2 (*n*; %)	3 (*n*; %)	4 (*n*; %)	5 (*n*; %)
Vaccination of people	208 (49.1)	6 (2.4)	9 (2.1)	13 (3.1)	47 (11.1)	141 (33.3)
Deworming of people	212 (50.0)	7 (1.7)	14 (3.3)	48 (11.3)	60 (14.2)	83 (19.6)
Primary hygiene care	203 (47.9)	5 (1.2)	2 (0.5)	7 (1.7)	31 (7.3)	176 (41.5)
Consumption of pasteurize milk	219 (51.7)	33 (7.8)	11 (2.6)	34 (8.0)	48 (11.3)	79 (18.6)
Vacuum the house frequently	208 (49.1)	2 (0.5)	21 (5.0)	50 (11.8)	61 (14.4)	82 (19.3)
Wear gloves when gardening	210 (49.5)	20 (4.7)	33 (7.8)	50 (11.8)	53 (12.5)	58 (13.7)
Frequent hand washing	205 (48.3)	4 (0.9)	1 (0.2)	11 (2.6)	38 (9.0)	165 (38.9)

^1^ 0: do not answer; 1: very little important; 2: little important; 3: important; 4: very important; 5: extremely important.

**Table 5 animals-11-03543-t005:** Level score attributed to the importance of each preventive measure directly associated with animals by respondents in northern Portugal in 2020 (*n* = 424).

Level of Importance ^1^	0 (*n*; %)	1 (*n*; %)	2 (*n*; %)	3 (*n*; %)	4 (*n*; %)	5 (*n*; %)
Vaccination of pets	202 (47.6)	3 (0.7)	1 (0.2)	4 (0.9)	18 (4.2)	196 (46.2)
Treatment of sick animals ^2^	205 (48.3)	1 (0.2)	2 (0.5)	1 (0.2)	17 (4.0)	198 (46.7)
Isolation/quarantine	212 (50.0)	13 (3.1)	8 (25.0)	25 (5.9)	40 (9.4)	126 (29.7)
Animal slaughter	229 (54.0)	63 (14.9)	34 (8.0)	39 (9.2)	32 (7.5)	27 (6.4)
Internal deworming of pets	205 (48.3)	4 (0.9)	4 (0.9)	11 (2.6)	42 (9.9)	158 (37.3)
External deworming of pets	207 (48.8)	2 (0.5)	2 (0.5)	9 (2.1)	43 (10.1)	161 (38.0)
Animal identification	207 (48.8)	14 (3.3)	7 (1.7)	27 (6.4)	44 (10.4)	125 (29.5)
Not feeding raw meat to pets ^3^	210 (49.5)	9 (2.1)	9 (2.1)	31 (7.3)	40 (9.4)	125 (29.5)

^1^ 0: do not answer; 1: very little important; 2: little important; 3: important; 4: very important; 5: extremely important. ^2^ *p* = 0.019 (higher importance for female owners). ^3^
*p* = 0.008 (higher importance for female owners).

**Table 6 animals-11-03543-t006:** Differences in the gender of respondents in northern Portugal in 2020 in relation to the level of importance given to different questions (*n* = 424).

Questions	Female Owners (*n* = 314)	Male Owners (*n* = 110)	*t*-Test	*p*-Value
Importance given to pets in the family	4.11 (± 1.15)	3.85 (± 1.08)	2.103	0.036
Importance given to vets in the protection of public health	3.48 (± 0.78)	3.10 (± 1.11)	3.94	0.000

Mean ± standard deviation.

## Data Availability

Not applicable.

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
