# Peer review of "A Cross-Sectional Study of Knowledge on Ownership, Zoonoses and Practices among Pet Owners in Northern Portugal"

_animals, 2021, doi:10.3390/ani11123543_

Round 1

Reviewer 1 Report

Dear authors,

the submitted paper is a good manuscript. There are interesting data and the subject is of great importance, also if it not a trend topic. 

The approach is good, it is very well written and you paid the right attention in the submission aspects and in the English language.

I have doubts about the methodology. It is not completely strong. Was the sample size calculated? Was the sample representative? Was the results analysed with a correct approach considering that it is a cross-sectional study?

If the design is appropriate, data are enough and well presented, but with a more detailed statistical approach, you can reach a higher level of quality of your paper.

I have some specific comments in the attached file.

Some of them are just suggestions, you can decide whether accept or not.

TITLE

I suggest you to change in “A cross-sectional survey” and not starting without an article

ABSTRACT

L 29: specify how the questionnaire was submitted (face-to-face?)

L 31: I think it is better to use “indoors” without “to”

INTRODUCTION

L 42: citations [1-3] are too reductive because you speak about difference among continents. You consider 2 papers from Europe and 1 from New Zealand. The ones from Europe only belong to UK and Ireland. You can probably be more precise if you cite a review like Downes BMC Vet Res 2013. You could also include Slater 2008 Prev Vet or Carvelli 2013 Biomed Res Int or Carvelli 2020 Plos One or Capello 2015 if you want to add an European perspective. And Baquero 2019 Brasil or Kisiel 2016 Mexico or Pulczer 2013 Guatemala or Leslie 1994 Canada if you want a global perspective. Probably, also the interesting Day 2016 (Callisto project) can be cited because the topic is very similar to your study. Not all of them, of course. The ones you like more, but with a wider perspective.

L 47: better “owned at least..:”. Isn’t it?

L 47: better “the most present” or similar instead of “top pet”?

GENERAL: why don’t cite also here which zoonoses you are speaking about?

L 74: why cite only papers dealing with immunocompromised people?

L 96-97: are you able to put a reference?

L 98: more data better than more questionnaire. Questionnaire is only the way you get the data

L 102: North capital letter

L 103: “epidemiological unit” better than “study target”?

L 106: are you able to put a reference?

M&M

L110: which part of Portugal? North of Portugal is not enough. Please define better the limit of the study area.

L 114: designed by who? Veterinarians? Social scientists?

L 115: how the short answer were categorised?

GENERAL:

  1. how do you calculate the sample size? With which assumptions (Confidence level, expected prevalence, degree of uncertainty?)
  2. questionnaire were administered in which way? Face-to-face? Where? How? You must have a design that choose a sample that is as more representative as possible of the general population.
  3. Questionnaire should be submitted as Supplementary Material both in Portuguese and English language.

RESULTS

It would be better to analyse the important data you got with a higher statistical analyses.

As this is a cross-sectional study, you should use a multivariable approach with a (Poisson) model and Prevalence ratios choosing for the desired outcome.

Table 1: why don’t you put the categories (Gender, age, education, occupation, residence) in a left column. In this way it is too confusing reading it.

Table 1: age classes are not well divided. Some range is only 2 years, some 4, some 20 and some 30. Use a statistical approach to define your class or choose the interval but with a minor difference among the classes.

Table 1: did you put the interquartile range you cite?

Table 1: I would suggest to use or the mean or the median. It’s a redundant information to use both as they are very similar.

GENERAL: you must specify the definition you used for categories: physical/mental activity and urban/rural residence.

Table 2: would it be interesting to differentiate dogs and cats in 2 different tables?

L 204: would not act what? Not clear to me.

L 207: what is treated? Sanitary approach?

Table 6:table has to be self-explicative. What is 4.11 and the other numbers?

DISCUSSION

L 360: the problem is not the small size. The problem is to understand and evaluate the representativeness of the sample size. 424 is not small if design and questionnaires are performed with a strong epidemiological approach.

L 391: the instead of de

L 398-407: too space for bovine and sheep brucellosis. Do you want to add something about B.canis?

GENERAL: the Discussion section is very long. It would be more “easy reading” if it was a bit shorter. Also if I don’t find something you could cut…However, why don’t you use paragraphs as in results?

CONCLUSIONS

L 607: …..revealed A lack of…..

L 607: add “North” Portuguese

L 611: I don’t think people need the ‘s

L 625: I would use the “eventual progress” or ”the trend” instead of “progress made”

TABLES and FIGURE

Captions need to be self-explicative. (Add in the North of Portugal in 2020 and so on)

Author Response

Dear Reviewer

Please find attached a revised version of our manuscript “Knowledge About Pet Ownership, Zoonoses and Practices of Pet Owners in the North of Portugal” [animals-1463201‏].

We would like to thank all the reviewers for their excellent suggestions that have led to the quality improvement of the manuscript. All the reviewer comments have been taken into consideration.

Questions by the reviewers (Q) and our answers (A) can be found below. A marked version of the manuscript (using track changes mode in MS Word) is being uploaded.

We hope that our manuscript may now be regarded as suitable for acceptance and publication in Animals as an Original Research.

Many thanks for your revision work.

REVIEWER #1:

  1. The submitted paper is a good manuscript. There are interesting data and the subject is of great importance, also if it not a trend topic.

The approach is good, it is very well written and you paid the right attention in the submission aspects and in the English language.

  1. A. Thank you for the constructive comments.

The manuscript has been revised taking into consideration the reviewer’s comments.

  1. I have doubts about the methodology. It is not completely strong. Was the sample size calculated? Was the sample representative? Was the results analysed with a correct approach considering that it is a cross-sectional study?

Being that:

n = size of the sample to be calculated

95% confidence level à Z = 1.96

e = maximum allowable margin of error (5%)

p = proportion expected to be found. The value of 50% (by default) was assumed.

  1. Q. If the design is appropriate, data are enough and well presented, but with a more detailed statistical approach, you can reach a higher level of quality of your paper.

  1. Reviewer 1 is right in her/his remark. We think we have now provided a more detailed statistical approach.

  1. Q. I have some specific comments in the attached file.

Some of them are just suggestions, you can decide whether accept or not.

  1. All specific comments and suggestions by Reviewer 1 are very much appreciated.

TITLE

  1. I suggest you to change in “A cross-sectional survey” and not starting without an article

  1. Thank you for the constructive comment. The title was changed to “Knowledge about pet ownership, zoonoses and practices of pet owners in the North of Portugal”.

ABSTRACT

  1. L 29: specify how the questionnaire was submitted (face-to-face?)

  1. Text has been updated to read as (please see line 29): A face-to-face questionnaire was completed by 424 pet owners, during November 2019 to February 2020.

  1. L 31: I think it is better to use “indoors” without “to”

  1. Text has been updated to read as (31): 73.1% allowed their pets free access indoors

INTRODUCTION

  1. L 42: citations [1-3] are too reductive because you speak about difference among continents. You consider 2 papers from Europe and 1 from New Zealand. The ones from Europe only belong to UK and Ireland. You can probably be more precise if you cite a review like Downes BMC Vet Res 2013. You could also include Slater 2008 Prev Vet or Carvelli 2013 Biomed Res Int or Carvelli 2020 Plos One or Capello 2015 if you want to add an European perspective. And Baquero 2019 Brasil or Kisiel 2016 Mexico or Pulczer 2013 Guatemala or Leslie 1994 Canada if you want a global perspective. Probably, also the interesting Day 2016 (Callisto project) can be cited because the topic is very similar to your study. Not all of them, of course. The ones you like more, but with a wider perspective.

  1. All the references cited by the reviewer have been consulted. Some of those references, together with others consulted by the authors, have now been cited and added to the reference list.

Added references:

Carvelli, A.; Iacoponi, F.; Scaramozzino, P. A cross-sectional survey to estimate the cat population and ownership profiles in a semirural area of Central Italy. Biomed Res. Int. 2016, 2016.

Kisiel, L.M.; Jones-Bitton, A.; Sargeant, J.M.; Coe, J.B.; Flockhart, D.T.T.; Reynoso Palomar, A.; Canales Vargas, E.J.; Greer, A.L. Owned dog ecology and demography in Villa de Tezontepec, Hidalgo, Mexico. Prev. Vet. Med. 2016, 135.

Baquero, O.S.; Queiroz, M.R. Size, spatial and household distribution, and rabies vaccination coverage of the Brazilian owned-dog population. Transbound. Emerg. Dis. 2019, 66, 1693–1700.

Carvelli, A.; Scaramozzino, P.; Iacoponi, F.; Condoleo, R.; Marta, U. Della. Size, demography, ownership profiles, and identification rate of the owned dog population in Central Italy. PLoS One. 2020, 15, 1–17.

  1. L 47: better “owned at least..:”. Isn’t it?

  1. Text has been updated to read as (lines 45-47): According to the European Pet Food Industry (FEDIAF), in 2019, 85 million European Union households (38% of all households) owned at least one pet.

  1. L 47: better “the most present” or similar instead of “top pet”?

  1. Text has been updated to read as (lines 47-48): Cats represent the most present pet, followed by dogs, birds, small mammals, ornamental fishes, and reptiles.

  1. GENERAL: why don’t cite also here which zoonoses you are speaking about?

  1. L 74: why cite only papers dealing with immunocompromised people?

  1. Not all studies cited refer to immunocompromised individuals. One study concerns a questionnaire with an approach similar to ours (Stull et al. 2012); another study corresponds to cancer patients (Chan et al. 2019); and one study refers to immunocompromised individuals (Mani et al. 2009).

  1. L 96-97: are you able to put a reference?

  1. Most of the articles (questionnaires that assess tutors’ knowledge about zoonoses) address a specific zoonosis, and not generalized knowledge about zoonoses. In addition, those articles refer to a single region. Therefore, there is no worldwide data on the subject, and even less follow-up studies to understand whether, or not, there has been an evolution in the acquisition of knowledge by tutors about zoonoses.

References have been added to the following sentence (lines 105-107): To date, few studies have assessed basic knowledge of pet-associated zoonoses among pet owners and, unfortunately, there are few longitudinal studies about this issue [20,33–38].

Added references:

Stull, J.W.; Peregrine, A.S.; Sargeant, J.M.; Weese, J.S. Household knowledge, attitudes and practices related to pet contact and associated zoonoses in Ontario, Canada. BMC Public Health 2012, 12.

Bingham, G.M.; Budke, C.M.; Slater, M.R. Knowledge and perceptions of dog-associated zoonoses: Brazos County, Texas, USA. Prev. Vet. Med. 2010, 93, 211–221.

Stull, J.W.; Peregrine, A.S.; Sargeant, J.M.; Weese, J.S. Pet husbandry and infection control practices related to zoonotic disease risks in Ontario, Canada. BMC Public Health 2013, 13.

Stull, J.W.; Brophy, J.; Sargeant, J.M.; Peregrine, A.S.; Lawson, M.L.; Ramphal, R.; Samson, L.; Bowes, J.; Weese, J.S. Knowledge, attitudes, and practices related to pet contact by immunocompromised children with cancer and immunocompetent children with diabetes. J. Pediatr. 2014, 165, 348-355.

Stull, J.W.; Brophy, J.; Weese, J.S. Reducing the risk of pet-associated zoonotic infections. CMAJ 2015, 187, 715–716.

Pereira, A.; Martins, Â.; Brancal, H.; Vilhena, H.; Silva, P.; Pimenta, P.; Diz-Lopes, D.; Neves, N.; Coimbra, M.; Alves, A.C.; et al. Parasitic zoonoses associated with dogs and cats: a survey of Portuguese pet owners’ awareness and deworming practices. Parasit. Vectors 2016, 9, 1–9.

Alho, A.M.; Lima, C.; Colella, V.; Madeira De Carvalho, L.; Otranto, D.; Cardoso, L. Awareness of zoonotic diseases and parasite control practices: a survey of dog and cat owners in Qatar. Parasit. Vectors 2018, 11, 1–7.

  1. L 98: more data better than more questionnaire. Questionnaire is only the way you get the data

  1. We understand the reviewer’s point of view. In the sentence (lines 107-110): “More questionnaires are needed with larger samples by region and country…”, we refer to that. Indeed, more data is needed, and we hope to contribute to that increase with future work.

  1. Q. L 102: North capital letter
  2. L 103: “epidemiological unit” better than “study target”?

  1. We have corrected the information. Text has been updated to read as (line 112): …using households with pets in the North of Portugal as epidemiological unit.

  1. L 106: are you able to put a reference?

  1. We thank the reviewer 1 for his/her suggestion. We have added two references that address the subject. References have been added to the following sentence (lines 112-116): Moreover, this study aimed to strengthen the message of combating zoonoses by promoting awareness among tutors and the general public, as well as underlining the importance of animal care and raising consciousness of the reality of abandonment of companion animals, often associated to ignorance on the topic [39,40].

Added references:

Cross, A.R.; Baldwin, V.M.; Roy, S.; Essex-Lopresti, A.E.; Prior, J.L.; Harmer, N.J. Zoonoses under our noses. Microbes Infect. 2019, 21, 10–19.

Prata, J.C. Strategies for the improvement of pet health and welfare in Portugal based on a pilot survey on husbandry, opinion, and information needs. Animals 2020, 10.

M&M

  1. L110: which part of Portugal? North of Portugal is not enough. Please define better the limit of the study area.

  1. A face-to-face questionnaire was applied in the North region of Portugal (in the districts of Vila Real, Braga and Oporto), to people who owned domestic animals and were willing to answer the questionnaire. Portugal is a medium-sized country on the European scale. The North region is the most populated region in Portugal. It represents 35% of the Portuguese population, which corresponds to around 3.6 M citizens; 22% of the total area of Continental Portugal; and 30% of the municipalities in mainland Portugal.

Text has been updated to read as (lines 119-124): A voluntary survey was conducted among pet owners from the North of Portugal (in the districts of Vila Real, Braga and Oporto). Portugal is a medium-sized country on the European scale. The North region is the most populated region in Portugal. It represents 35% of the Portuguese population, which corresponds to around 3.6 M of citizens; 22% of the total area of Continental Portugal; and 30% of the municipalities in mainland Portugal [41].

Added reference:

INE - Plataforma de divulgação dos Census 2021. Resultados preliminares. Available online: https://www.ine.pt/scripts/db_censos_2021.html (accessed on 25 November 2021).

  1. Q. L 114: designed by who? Veterinarians? Social scientists?

  1. The questionnaire was designed by veterinarians and epidemiologists. Text has been updated to read as (lines 132-134): The questionnaire was based on a literature review and designed by the authors (veterinarians and epidemiologists) to obtain information about zoonoses and pet ownership.

  1. Q. L 115: how the short answer were categorised?

  1. Whenever possible, short answers were categorized the simplest way, i.e. yes or no. Other short answers were the name of diseases/zoonoses. Text has been updated to read as (lines 134-135): The questionnaire consisted of closed or short answer (which can be categorized) questions.

GENERAL:

  1. Q. 1. how do you calculate the sample size? With which assumptions (Confidence level, expected prevalence, degree of uncertainty?)

  1. The sample size was calculated using the following formula:

Being that:

n = size of the sample to be calculated

Z = deviation from the mean value accepted to achieve the desired confidence level. Depending on the desired level of confidence, a certain value is used, which is given by the shape of the Gaussian distribution. The most frequent values are:

95% confidence level à Z = 1.96

e = maximum allowable margin of error (5%)

p = proportion expected to be found. The value of 50% (by default) was assumed.

Text has been added to read as (lines 128-132): The sample size of this study was calculated according to the formula of survey sample size calculation [42], assuming 50% of pet owners with sufficient knowledge and practices. Therefore, 384 participants were required. To allow for a 10% non-response, the smallest sample size required was 422 participants.

Added reference:

Hajian-Tilaki, K. Sample size estimation in epidemiologic studies. Casp. J. Intern. Med. 2011, 2, 289–298.

  1. 2. questionnaire were administered in which way? Face-to-face? Where? How? You must have a design that choose a sample that is as more representative as possible of the general population.

  1. A face-to-face questionnaire was applied in the North region of Portugal (in the districts of Vila Real, Braga and Oporto.), to people who owned domestic animals and were willing to answer the questionnaire. Portugal is a medium-sized country on the European scale. The North region is the most populated region in Portugal. It represents 35% of the Portuguese population, which corresponds around to 3.6 M citizens; 22% of the total area of Continental Portugal; and 30% of the municipalities in mainland Portugal.

We started this work in the North region because it is representative of the general population, and it is the area of residence of all team members. Nevertheless, we consider it important to extend our study to other regions of the country in the future.

Text has been updated to read as (lines 119-124): A voluntary survey was conducted among pet owners from the North of Portugal (in the districts of Vila Real, Braga and Oporto). Portugal is a medium-sized country on the European scale. The North region is the most populated region in Portugal. It represents 35% of the Portuguese population, which corresponds to around 3.6 M of citizens; 22% of the total area of Continental Portugal; and 30% of the municipalities in mainland Portugal [41].

Added reference:

INE - Plataforma de divulgação dos Census 2021. Resultados preliminares. Available online: https://www.ine.pt/scripts/db_censos_2021.html (accessed on 25 November 2021).

  1. 3. Questionnaire should be submitted as Supplementary Material both in Portuguese and English language.

  1. the suggestion by Reviewer 1 is very much appreciated. The questionnaire will be submitted as Supplementary Material both in Portuguese and English.

Text has been updated to read as (lines 126-128): After agreeing to participate in the study, everyone was asked to complete an anonymous, confidential 15-minute self-administered written questionnaire on-site.

RESULTS

  1. It would be better to analyse the important data you got with a higher statistical analyses.

  1. the suggestion by Reviewer 1 is very much appreciated. We have provided a complementary statistical analysis. Nevertheless, it has not been our purpose to make this study a statistical exercise.

  1. As this is a cross-sectional study, you should use a multivariable approach with a (Poisson) model and Prevalence ratios choosing for the desired outcome.

  1. The aim of the study was to gather information regarding participants’ demographic characteristics; types of pets in the household and their importance to the family; pet contact-related attitudes; knowledge about zoonoses, high-risk groups, pathways of disease transmission and disease protection measures. The aim of the study was not to collect information about the risk factors associated with pet ownership, as such, the study was not designed with the aim of performing a multivariate analysis. Of course, that study design would be very interesting, but it would result in an article totally different from the present one, which was not intended with this approach. However, we intend to do that in future studies.

  1. Table 1: why don’t you put the categories (Gender, age, education, occupation, residence) in a left column. In this way it is too confusing reading it.

  1. We have reformulated Table 1.

  1. Table 1: age classes are not well divided. Some range is only 2 years, some 4, some 20 and some 30. Use a statistical approach to define your class or choose the interval but with a minor difference among the classes.

  1. The parameter “age” in Table 1 has been changed. We have divided it in two classes: under 35 years old and above 35 years old. The reason is that 35 years of age is considered the threshold between young adults and adults. And it is very close to the average age observed in the study.

  1. Q. Table 1: did you put the interquartile range you cite?

  1. The interquartile range was eliminated due to the changes made.

  1. Table 1: I would suggest to use or the mean or the median. It’s a redundant information to use both as they are very similar.

  1. The Reviewer is right in her/his remark, and we thank her/him very much for pointing out this important detail. We deleted the mean and only inserted the median.

  1. GENERAL: you must specify the definition you used for categories: physical/mental activity and urban/rural residence.

  1. We divided the profession into physical labor and mental labor to reduce the degrees of freedom and, consequently facilitate statistical analysis. We have based the definition of these categories on common sense. In fact, the distinction of physical and mental activities has seemed obvious from the received answers, with the same being true for urban or rural residence.

  1. Table 2: would it be interesting to differentiate dogs and cats in 2 different tables?

  1. The suggestion by Reviewer 1 is very much appreciated. However, it was not the aim of the study. The questionnaire was not built to make this separation of variables.

  1. Q. L 204: would not act what? Not clear to me.

  1. 31.3% of respondents answered that they would not take any action if a dog or cat appeared near their area of residence. This means they would not help the animal at all, and they would not try to resolve the situation.

Text has been updated to read as (lines 234-236): In this study, 31.3% of the participants would not act, which means they would not help the animal or try to solve the situation.

  1. L 207: what is treated? Sanitary approach?

  1. Most respondents considered that stray dogs should be collected, vaccinated, dewormed, neutered. They also considered that other treatments (surgical and/or medical treatment) should be applied when necessary.

Text has been updated to read as (lines 239-241): The majority (85.4%; n = 362) of the respondents considered that stray dogs should be collected and treated (surgical and/or medical treatment should be applied when necessary).

  1. Table 6: table has to be self-explicative. What is 4.11 and the other numbers?

  1. We have reformulated Table 6.

DISCUSSION

  1. L 360: the problem is not the small size. The problem is to understand and evaluate the representativeness of the sample size. 424 is not small if design and questionnaires are performed with a strong epidemiological approach.

  1. Reviewer 1 poses an important question. Nevertheless, we think that we have already provided enough evidence that we followed procedures to assure that the sample is representative.

  1. L 391: the instead of de

  1. Text has been updated to read as (line 432): Pet owners who claimed to know the term “zoonoses” were asked for examples.

  1. L 398-407: too space for bovine and sheep brucellosis. Do you want to add something about B.canis?

  1. The paragraph was deleted.

  1. GENERAL: the Discussion section is very long. It would be more “easy reading” if it was a bit shorter. Also if I don’t find something you could cut…However, why don’t you use paragraphs as in results?

  1. We have reformulated the Discussion.

CONCLUSIONS

  1. L 607: …..revealed A lack of…..

L 607: add “North” Portuguese

  1. Text has been updated to read as (lines 666-667): This study revealed a lack of knowledge and understanding of zoonoses among North Portuguese pet owners.

  1. L 611: I don’t think people need the ‘s
  2. Text has been updated to read as (lines 669-670): Different ways should be used to draw people attention to the topic…

  1. L 625: I would use the “eventual progress” or “the trend” instead of “progress made”

  1. Text has been updated to read as (lines 684-685): …should be developed to frequently update the eventual progress.

TABLES and FIGURE

  1. Q. Captions need to be self-explicative. (Add in the North of Portugal in 2020 and so on)

  1. Captions have been updated to read as:

Table 1. Demographics of respondents who participated in the survey in the North of Portugal in 2020 (n = 424).

Table 2. High-risk practices for the transmission of zoonotic diseases among respondents in the North of Portugal in 2020 (n = 424).

Table 3. Risk score attributed to each high-risk group by respondents in the North of Portugal in 2020 (n = 424).

Table 4. Level score attributed to the importance of each general preventive measure by respondents in the North of Portugal in 2020 (n = 424).

Table 5. Level score attributed to the importance of each preventive measure directly associated to animals by respondents in the North of Portugal in 2020 (n = 424).

Table 6. Differences in the gender of respondents in the North of Portugal in 2020 in relation to the level of importance given to different questions (n = 424).

Reviewer 2 Report

Abstract line 32 – assumed they did it daily – not clear what that refers to, the pet sleeping in the bed?

The abstract lacks to bring together the results as in risk behaviours and knowledge clearly referred to – as for now I assume sleeping in owners’ beds and licking/kissing are risk behaviours. This could be strengthened.

Introduction

I don’t quite see the need yet for such a study if the risk of zoonotic diseases is very low. Why is it important to study this? Is there a relation to animal health or welfare? The author mentioned One Health in the abstract, could they include this conceptual framework in the introduction to make clear why this study is important?

Methods

How was the study site selected, why North Portugal? How was the sampling conducted?

Did the authors ask sex or gender?

Data analysis – the authors present categorical data collection tools – but present data analysis for continuous scales e.g. t-test, this should only be used when using likert-type scales (where only end-points are defined and numbers are used in between as this sets the assumption of equal distances, if words are used to describe the single categories equal distances between categories cannot be assumed)

It is not clear what was asked and how the questions of the questionnaire were analysed. Did the authors use single items for the analysis or were scales used? What were the actual questions?

The questions regarding zoonosis, were they open ended? Or could participants choose from a list? It is not clear how this was assessed.

It is not clear how the constructs presented in the results were evaluated. I gave some examples but also risk and who is in the risk group etc. how did participants answer these questions. The methods part should be written in a way that I can replicate the whole study by just reading this part.

Results

3.2. How were benefits and risks assessed? It is not clear how the benefits outweigh the risks. What test was used comparing male and female participants and what was the measure that was compared? Single item?

Please add test-statistics if you use chi-square please add the value and df. It is otherwise not clear what was compared.

Not clear from methods that attitudes towards stray animals were assessed. Why is that important?

It is difficult to judge the validity of the results, since there are no0 test statistics and it is not clear how the questions were asked.

Discussion

I am not sure you can call it household knowledge, you asked a single person or was the questionnaire answered by multiple people in the household?

In general, the discussion and introduction do not fit. It reads like zoonotic diseases are very prevalent in Portugal, but the introduction says it is rare. There is a lot of emphasis on single diseases even when they are prevalent in livestock. Did participants just mention what they knew regardless of where it occurs? I am not sure how to interpret the results, mainly because it is not clear how questions were asked. If participants were asked about potential risks then they may assume there are risks involved, there is a lot of literature on that. Also if they are asked about disease prevalence they may assume it must be prevalent otherwise you wouldn’t have asked. These biases need to be discussed. But foremost the reader needs to know what was actually asked so it is possible to validate the results and understand the discussion.

Author Response

Dear Reviewer

Please find attached a revised version of our manuscript “Knowledge About Pet Ownership, Zoonoses and Practices of Pet Owners in the North of Portugal” [animals-1463201‏].

We would like to thank all the reviewers for their excellent suggestions that have led to the quality improvement of the manuscript. All the reviewer comments have been taken into consideration.

Questions by the reviewers (Q) and our answers (A) can be found below. A marked version of the manuscript (using track changes mode in MS Word) is being uploaded.

We hope that our manuscript may now be regarded as suitable for acceptance and publication in Animals as an Original Research.

Many thanks for your revision work.

REVIEWER # 2

  1. Thank you for the constructive comments.

The manuscript has been revised taking into consideration the reviewer’s comments.

  1. Abstract line 32 – assumed they did it daily – not clear what that refers to, the pet sleeping in the bed?

  1. We thank Reviewer 2 for her/his useful comments and suggestions. Regarding this particular comment, 29% of the respondents admitted that they allow their pets to sleep in the bed daily.

  1. The abstract lacks to bring together the results as in risk behaviours and knowledge clearly referred to – as for now I assume sleeping in owners’ beds and licking/kissing are risk behaviours. This could be strengthened.

  1. We thank Reviewer 2 for her/his useful comments. It is not possible for the abstract to describe the entire work, due to the limited number of words. Nevertheless, we are convinced that the abstract is enough to attract and direct the reader to the essentials of main text. We think the following sentence in the abstract can be enlightening for the reviewer (lines 35-38): “The low level of knowledge of pet owners and the occurrence of high-risk behaviors indicate a need to strengthen communication between veterinarians, physicians, pet owners and the general public to reduce the risk of acquisition and transmission of zoonoses”.

Introduction

  1. I don’t quite see the need yet for such a study if the risk of zoonotic diseases is very low. Why is it important to study this? Is there a relation to animal health or welfare? The author mentioned One Health in the abstract, could they include this conceptual framework in the introduction to make clear why this study is important?

  1. We apologize for any potential misunderstanding. Text has been updated to read as (lines 69-70): The number of known emerging infections from companion animals is low [18]. However, there is potential …

It is important to investigate the weaknesses / lack of knowledge of tutors (and the general population) regarding zoonoses, since many of the disease risks related to pet ownership and pet contact can be eliminated or reduced through simple measures (proper pet husbandry, hand hygiene, altered animal-contact behaviors). Studying this subject is the first step to clarify the job that needs to be done to promote awareness and education among tutors and general population and, subsequently, effective disease prevention.

Text added (lines 97-104): The One Health concept has become an international standard for expanding interdisciplinary collaboration between professionals in all aspects of human, animal and environmental health, working at the local, regional, national and global levels [29]. The adoption of a One Health strategy and mentality is imperative to predict, prevent, detect at early stages, respond and control infectious diseases [30]. Therefore, engaging communities (pet owners and general population) through education and awareness about zoonotic diseases and their prevention is a fundamental part of this concept to improve human and animal health and welfare [31,32].

Added references:

Benelli, G.; Duggan, M.F. Management of arthropod vector data – social and ecological dynamics facing the One Health perspective. Acta Trop. 2018, 182, 80–91.

Machalaba, C.C.; Salerno, R.H.; Barton Behravesh, C.; Benigno, S.; Berthe, F.C.J.; Chungong, S.; Duale, S.; Echalar, R.; Karesh, W.B.; Ormel, H.J.; et al. Institutionalizing One Health: from assessment to action. Heal. Secur. 2018, 16, 37–43.

Cunningham, A.A.; Daszak, P.; Wood, J.L.N. One Health, emerging infectious diseases and wildlife: two decades of progress? Philos. Trans. R. Soc. B Biol. Sci. 2017, 372.

van Herten, J.; Bovenkerk, B.; Verweij, M. One Health as a moral dilemma: towards a socially responsible zoonotic disease control. Zoonoses Public Health 2019, 66, 26–34.

Methods

  1. How was the study site selected, why North Portugal? How was the sampling conducted?

  1. A face-to-face questionnaire was applied in the North region of Portugal (in the districts of Vila Real, Braga and Oporto.), to people who owned domestic animals and were willing to answer the questionnaire. Portugal is a medium-sized country on the European scale. The North region is the most populated region in Portugal. It represents 35% of the Portuguese population, which corresponds around to 3.6 M citizens; 22% of the total area of Continental Portugal; and 30% of the municipalities in mainland Portugal.

We started this work in the North region because it is representative of the general population, and it is the area of residence of all team members. Nevertheless, we consider it important to extend our study to other regions of the country in the future.

  1. Did the authors ask sex or gender?

  1. Gender.

  2. Data analysis – the authors present categorical data collection tools – but present data analysis for continuous scales e.g. t-test, this should only be used when using likert-type scales (where only end-points are defined and numbers are used in between as this sets the assumption of equal distances, if words are used to describe the single categories equal distances between categories cannot be assumed)

  1. A survey scale with a set of answer options – numeric and verbal – covering a range of opinions on some topics was used to measure opinions in a reliable way. As example, participants who responded positively to the question about the importance of pets in the family were asked the level of importance, given that for each verbal option was assigned a numerical value. Thus:

1: very little important

2: not very important

3: important

4: very important

5: extremely important

  1. It is not clear what was asked and how the questions of the questionnaire were analysed. Did the authors use single items for the analysis or were scales used? What were the actual questions?

  1. The questionnaire was included as supplement material to clarify some potential misunderstandings.

  1. The questions regarding zoonosis, were they open ended? Or could participants choose from a list? It is not clear how this was assessed.

  1. The questionnaire was included as supplement material to clarify some potential misunderstandings.

  1. It is not clear how the constructs presented in the results were evaluated. I gave some examples but also risk and who is in the risk group etc. how did participants answer these questions. The methods part should be written in a way that I can replicate the whole study by just reading this part.

  1. Reviewer 2 is right in her/his remark. However, we think that with the presentation of the questionnaires, those aspects may have become clearer.

Results

  1. 3.2. How were benefits and risks assessed? It is not clear how the benefits outweigh the risks. What test was used comparing male and female participants and what was the measure that was compared? Single item?

  1. We did not study or assess the risks and benefits. That was not the purpose of the study. We only intended to determine whether respondents were aware of the common/general benefits and risks of owning an animal.

Benefits – pets as companions; prevention of loneliness; promotion of physical activity; creation of habits and routines; foster a sense of responsibility; reduction of stress levels.

Risks – diseases and infections; bites and scratches and potential negative psychological effects.

The benefits outweigh the risks when the individual/respondent considers that having a pet is more rewarding than the potential negative aspects of having it.

  1. Please add test-statistics if you use chi-square please add the value and df. It is otherwise not clear what was compared.

  1. Values and df have been added in the manuscript.

  1. Not clear from methods that attitudes towards stray animals were assessed. Why is that important?

  1. The attitude that the respondent would have towards a stray animal was assessed. We wanted to know if the respondent would: a) provide basic care (food, shelter); b) take the animal to the veterinarian for a check-up and medical needs; c) communicate to the responsible entities (Municipal Council, city kennel) or to non-governmental entities (veterinary hospitals/clinics, animal protection associations); d) not take any action.

This question is important to understand if the respondents would feel compelled to help the animal and, as citizens, if they would take the responsibility to try to solve this public health problem.

Text added (lines 141-144): …pet and animal contact-related attitudes (provide basic care - food, shelter; take the animal to the veterinarian; communicate to the responsible entities or to non-governmental entities; not take any action) …

  1. It is difficult to judge the validity of the results, since there are no0 test statistics and it is not clear how the questions were asked.

  1. The questionnaires have been added as supplements. All statistic information has been added.

Discussion

  1. I am not sure you can call it household knowledge, you asked a single person or was the questionnaire answered by multiple people in the household?

  1. The study that served as an example for the development of our questionnaire (Stull et al. 2012) is entitled: “Household knowledge, attitudes and practices related to pet contact and associated zoonoses in Ontario, Canada”. In that study, the questionnaire was answered only by one member of each household with whom the team came into contact: “A single member from each household was asked to complete an anonymous, confidential 10-minute self-administered written questionnaire on-site or until later time”. The fact that the article by Stull et al., which is one of the most cited and used as an example in this matter, has used the expression “household knowledge” does not seem inappropriate to us. In addition, many of our questions are related to the family environment, so we do not understand the possible inconvenience in using such an expression.

Text has been updated to read as (lines 126-128): After agreeing to participate in the study, a single member of each household was asked to complete an anonymous, confidential 15-minute self-administered written questionnaire on-site.

  1. In general, the discussion and introduction do not fit. It reads like zoonotic diseases are very prevalent in Portugal, but the introduction says it is rare.

  1. This misunderstanding is due to a poorly formulated sentence, but already corrected (lines 69-70).

  1. There is a lot of emphasis on single diseases even when they are prevalent in livestock. Did participants just mention what they knew regardless of where it occurs?

  1. Participants were asked for examples of zoonoses. The focus of the question was to assess whether the participants who answered knowing the definition of zoonoses really knew it by giving examples.

  1. I am not sure how to interpret the results, mainly because it is not clear how questions were asked. If participants were asked about potential risks then they may assume there are risks involved, there is a lot of literature on that. Also if they are asked about disease prevalence they may assume it must be prevalent otherwise you wouldn’t have asked. These biases need to be discussed. But foremost the reader needs to know what was actually asked so it is possible to validate the results and understand the discussion.

  1. The questionnaire has been added for further clarification. There may be some less well understood detail resulting from the translation/retroversion into English.

Round 2

Reviewer 1 Report

Dear authors,

The revised manuscript was improved. I still believe that with these good data collected, the output of your work could have been greater from an epidemiological point of view. Anyhow, it’s your choice. And I really believe that this activity is among peers.

There are only few changes to be addressed in my opinion.

M&M

L125 -131: where did you get the interviewers? Vet practices, postal office, sending questionnaires by email (in case where does email list come from?) It is important to state it to make readers aware that you choose a sample that is as more representative as possible of the general population. Unless, it is not a problem to state that you used a convenience sampling, but in this case, you should state it.

L 134: I think that it is more elegant and more appropriate to state that an unknown prevalence has been set. Unless you really set the prevalence of the outcome variable as 50%. This will not change the sample size.

GENERAL: I suggest to add: “the definitions of urban/rural residence and physical/mental occupation (are there others?) were not present in the questionnaire, but they were based on respondents’ definition”.

RESULTS

GENERAL: I would suggest removing the degree of freedom from the text…

Author Response

Dear Reviewer

Please find attached a revised version of our manuscript “Knowledge About Pet Ownership, Zoonoses and Practices of Pet Owners in the North of Portugal” [animals-1463201‏].

We would like to thank all the reviewers for their excellent suggestions that have led to the quality improvement of the manuscript. All the reviewer comments have been taken into consideration.

Questions by the reviewers (Q) and our answers (A) can be found below. A marked version of the manuscript (using track changes mode in MS Word) is being uploaded.

We hope that our manuscript may now be regarded as suitable for acceptance and publication in Animals as an Original Research.

Many thanks for your editorial work.

With kind regards,

Prof. Ana Cláudia Coelho

REVIEWER #1:

  1. The revised manuscript was improved. I still believe that with these good data collected, the output of your work could have been greater from an epidemiological point of view. Anyhow, it’s your choice. And I really believe that this activity is among peers.

There are only few changes to be addressed in my opinion.

  1. A. Thank you for the constructive comments.

The manuscript has been revised taking into consideration the reviewer’s comments.

M&M

  1. L125 -131: where did you get the interviewers? Vet practices, postal office, sending questionnaires by email (in case where does email list come from?) It is important to state it to make readers aware that you choose a sample that is as more representative as possible of the general population. Unless, it is not a problem to state that you used a convenience sampling, but in this case, you should state it.
  2. Reviewer 1 is right in her/his remark. We apologize for not being more explicit. We used a convenience sampling.

Text has been updated to read as (lines 119-120): A voluntary survey was conducted among a convenience sample of pet owners from the North of Portugal …

  1. L 134: I think that it is more elegant and more appropriate to state that an unknown prevalence has been set. Unless you really set the prevalence of the outcome variable as 50%. This will not change the sample size.
  2. Text has been updated to read as (lines 128-132): The sample size of this study was calculated according to the formula of survey sample size calculation [42]. Assuming a 50% default prevalence, a 95% confidence level, and a 10% absolute error, 384 participants were necessary to include in the study. To allow for a 10% non-response, the smallest sample size required was 422 participants.
  3. GENERAL: I suggest to add: “the definitions of urban/rural residence and physical/mental occupation (are there others?) were not present in the questionnaire, but they were based on respondents’ definition”.
  4. Text has been updated to read as (lines 140-143): … respondents’ demographic characteristics (gender, age, education level, residence, and occupation). The definitions of urban/rural residence and physical/mental occupation were not present in the questionnaire, but they were based on respondent’s definition.

RESULTS

  1. GENERAL: I would suggest removing the degree of freedom from the text…
  2. Degrees of freedom have been removed from the text.

Reviewer 2 Report

Thank you for addressing the comments, it is a lot clearer now especially with regards to the methods. One comment though just because one study that may have been cited a lot for certain aspects refers to household knowledge does not mean that you can refer to it as household knowledge - it is individual knowledge after all as you haven't tested each household member. Living together does not mean sharing all the knowledge and there are plenty of studies out that show that the one who cares for the pet most possess greater knowledge.

Author Response

Dear Reviewer

Please find attached a revised version of our manuscript “Knowledge About Pet Ownership, Zoonoses and Practices of Pet Owners in the North of Portugal” [animals-1463201‏].

We would like to thank  for Your excellent suggestions that have led to the quality improvement of the manuscript. All the reviewer comments have been taken into consideration.

We hope that our manuscript may now be regarded as suitable for acceptance and publication in Animals as an Original Research.

Many thanks for your editorial work.

With kind regards,

Prof. Ana Cláudia Coelho

REVIEWER #2:

  1. Thank you for addressing the comments, it is a lot clearer now especially with regards to the methods. One comment though just because one study that may have been cited a lot for certain aspects refers to household knowledge does not mean that you can refer to it as household knowledge - it is individual knowledge after all as you haven't tested each household member. Living together does not mean sharing all the knowledge and there are plenty of studies out that show that the one who cares for the pet most possess greater knowledge.
  2. Thank you for the constructive comments.